# Active Assessment of Prediction Services as Accuracy Surface Over Attribute Combinations

**Vihari Piratla**[*]    **Soumen Chakrabarty**    **Sunita Sarawagi**
Department of Computer Science
Indian Institute of Technology, Bombay

## Abstract

Our goal is to evaluate the accuracy of a black-box classification model, not as a single aggregate on a given test data distribution, but as a surface over a large number of combinations of attributes characterizing multiple test data distributions. Such attributed accuracy measures become important as machine learning models get deployed as a service, where the training data distribution is hidden from clients, and different clients may be interested in diverse regions of the data distribution. We present Attributed Accuracy Assay (AAA) — a Gaussian Process (GP)-based probabilistic estimator for such an accuracy surface. Each attribute combination, called an 'arm', is associated with a Beta density from which the service's accuracy is sampled. We expect the GP to smooth the parameters of the Beta density over related arms to mitigate sparsity. We show that obvious application of GPs cannot address the challenge of heteroscedastic uncertainty over a huge attribute space that is sparsely and unevenly populated. In response, we present two enhancements: pooling sparse observations, and regularizing the scale parameter of the Beta densities. After introducing these innovations, we establish the effectiveness of AAA in terms of both its estimation accuracy and exploration efficiency, through extensive experiments and analysis. Our code and dataset can be found at: https://github.com/vihari/AAA.

## 1 Introduction

Increasing concentration of big data and computing resources has resulted in widespread adoption of machine learning as a service (MLaaS). The best-performing NLP, speech, image and video recognition tools are now provided as network services. MLaaS comes with few accuracy specifications or service level agreements, perhaps only leaderboard numbers from benchmarks that may not be closely related to most clients' deployment data distributions. The client, therefore, finds it difficult to choose the best provider without extensive pilot trials [1]. Different clients may need to deploy the service on very different data distributions, with possibly widely different accuracy.

In such circumstances, we propose that a service provider, or a service standardization agency, publish the accuracy of the classifier, not as one or few aggregate numbers, but as a *surface* defined on a space of input instance *attributes* that capture the variability of consumer expectations. Indoor/outdoor, day/night, urban/rural may be attributes of input images for visual object recognition tasks. Speaker age, gender, ethnicity/accent may be attributes of input audio for speech recognition tasks. We call a combination of attributes in their Cartesian space an *arm* (borrowing from bandit terminology)[2]. The labeled instances used by the service provider may not represent or cover well the space of attributes of interest to subscribers. Labeled data may be proprietary and inaccessible to prospective consumers and standardization agencies. Whoever estimates the accuracy surface, therefore, needs to

---

[*]vihari@cse.iitb.ac.in

[2]Figure 1 shows an example of diverse accuracy over arms.

35th Conference on Neural Information Processing Systems (NeurIPS 2021).

*actively* select instances from an unlabeled pool for labeling, presumably within a restricted budget, to adequately cover the attribute space.

Several recent studies have highlighted the variability in accuracy across data sub-populations [2, 3], specifically in the context of fairness [4, 5, 6], and also proposed active estimation techniques of sub-population accuracy [7, 8]. We solve a more general problem where the space of arms (sub-population) defined by the Cartesian space of attributes grows combinatorially. This inevitably leads to extreme sparsity of labeled instances for many arms. A central challenge is how to smooth the estimate across related arms while faithfully representing the uncertainty for active exploration.

We present Attributed Accuracy Assay (AAA) — a practical system that estimates accuracy, together with the uncertainty of the estimate, as a function of the attribute space. AAA uses these estimates to drive the sampling policy for each attribute combination. Gaussian Process (GP) regression is a natural choice to obtain smooth probabilistic accuracy estimates over arm attributes. However, a straightforward GP model fails to address the challenge of heteroscedasticity that we face with uneven and sparse supervision across arms. We model arm-specific service accuracy as drawn from a Beta density that is characterized by mean and scale parameters, which are sampled from two GPs that are informed by suitable trained kernels over the attribute space. We propose two further enhancements to the training of this model. First, we recognize an over-smoothing problem with GP's estimation of the Beta scale parameters, and propose a Dirichlet likelihood to supervise the relative values of scale across arms. Second, we recognize that arms with very low support interfere with learning the kernel parameters of the GPs. We mitigate this by pooling observations across related arms. With these fixes, AAA achieves the best estimation performance among competitive alternatives.

Another practical challenge in our setting is that some attributes of instances are not known exactly. For example, attributes, such as camera shutter speed or speaker gender, may be explicitly provided as meta information attached with instances. But other attributes, such as indoor/outdoor, or speaker age, may have to be estimated noisily via another (attribute) classifier, because accurate human-based acquisition of attributes would be burdensome. AAA also tackles uncertain attribute inference. Its attribute classifiers are trained on a small amount of labeled data and their error rates are modeled in a probabilistic framework.

We report on extensive experiments using several real data sets. Comparison with several estimators based on Bernoulli arm parameters, Beta densities per arm, and even simpler forms of GPs on the arm Beta distributions, shows that AAA is superior at quickly cutting down arm accuracy uncertainty.

Summarizing, our contributions are:

- We motivate and define the problem of accuracy surface estimation over a large space of attribute combinations.
- Our proposed estimator AAA fits a Beta density for every attribute combination (arm), with its parameters smoothed via two GPs to capture heteroscedastic uncertainty of each arm's accuracy under limited data settings.
- We propose two important components included in AAA: 1) a Dirichlet regularization to control over-smoothing of the Beta scale parameters, and 2) pooled observations to reduce over-fitting of a GP-associated kernel to sparse arms.
- We show significant gains in terms of both estimation quality and the efficiency of exploration on four real classification models compared to existing methods. AAA obtains an average 80% reduction in macro averaged square error over the existing methods.

## 2   Problem Setup

Our goal is to evaluate a given machine learning service model $S$ used by a diverse set of consumers. The service $S : \mathcal{X} \mapsto \mathcal{Y}$ could be any predictive model that, for an input instance $\mathbf{x} \in \mathcal{X}$, assigns an output label $\hat{y} \in \mathcal{Y}$, where $\mathcal{Y}$ is a discrete label space. Let $y(\mathbf{x})$ denote the true label of $\mathbf{x}$ and $\text{Agree}(y, \hat{y})$ denote the match between the two labels. For scalar classification, $\text{Agree}(y, \hat{y})$ is in {0,1}. For structured outputs, e.g., sequences, we could use measures like BLEU scores in [0,1]. Classifiers are routinely evaluated on their expected accuracy on a data distribution $P(\mathcal{X}, \mathcal{Y})$:

$$\rho = \mathbb{E}_{P(\mathbf{x}, y)}[\text{Agree}(y, S(\mathbf{x})] \tag{1}$$

We propose to go beyond this single measure and define accuracy as a surface over a space of attributes of the input instances. Let $A$ denote a list of $K$ attributes that capture the variability of

consumer expectation on which the service $S$ will be deployed. For instance, visual object recognition is affected by the background scene, and facial recognition is affected by demographic attributes. We use $A(\mathbf{x}) \in \mathcal{A}$ to denote the vector of values of attributes of input $\mathbf{x}$ and $\mathcal{A}$ to denote the Cartesian product of the domains of all attributes. An attribute could be discrete, e.g., the ethnicity of a speaker; Boolean, e.g., whether a scene is outdoors/indoors; or continuous, e.g., the age of the speaker in speech recognition. Some of the attributes of $\mathbf{x}$, for example the camera settings of an image, may be known exactly, and others may only be available as a distribution $M_k(a_k|\mathbf{x})$ for an attribute $a_k \in A$, obtained from a pre-trained probabilistic classifier.

Generalizing from a single global expected accuracy (1), we define the accuracy surface $\rho : \mathcal{A} \to [0, 1]$ of a service $S$ at each attribute combination $\boldsymbol{a} \in \mathcal{A}$, given a data distribution $P(\mathcal{X}, \mathcal{Y})$, as

$$\rho(\boldsymbol{a}) = \mathbb{E}_{P(\mathbf{x}, y | A(\mathbf{x}) = \boldsymbol{a})}[\text{Agree}(y, S(\mathbf{x}))] \tag{2}$$

Our goal is to provide an estimate of $\rho(\boldsymbol{a})$ given two kind of data sampled from $P(\mathcal{X}, \mathcal{Y})$: a small labeled sample $D$, and a large unlabeled sample $U$. In addition, we are given a budget of $B$ instances for which we can seek labels $y$ from a human by selecting them from $U$. Applying $M_k$ to all of $U$ is, however, free of cost.

We aim to design a probabilistic estimator for $\rho(\boldsymbol{a})$, which we denote as $P(\rho|\boldsymbol{a})$ where $\rho \in [0, 1]$ and $\boldsymbol{a} \in \mathcal{A}$. This is distinct from active learning, which selects instances to train the learner toward greater accuracy, and also active accuracy estimation [7], which does not involve a surface over $\boldsymbol{a}$s. We also show that standard tools to regress from $\boldsymbol{a}$ to $\rho$ are worse than our proposal.

We measure the quality of our estimate as the square error between the gold accuracy $\rho(\boldsymbol{a})$ and the mean of the estimated accuracy distribution $P(\rho|\boldsymbol{a})$. Our estimator distribution naturally gives an idea of the posterior variance of accuracy estimate of each attribute combination, which we use for uncertainty-based exploration.

## 3 Proposed Estimator

We will first review recent work that leads to candidate solutions to our problem, discuss their limitations, and finally present our solution. Initially, to keep the treatment simple, we assume $A(\mathbf{x})$ and gold $y$ (hence $c = \text{Agree}(S(x), y)$, the service correctness bit) is known for all instances. Later in this section, we remove these assumptions.

The simplest option is to ignore any relationship between arms, and, for each arm $\boldsymbol{a}$, fit a suitable density over $\rho(\boldsymbol{a})$. When this density is sampled, we get a number in $[0, 1]$, which is like a coin head probability used to sample correctness bits $c$. For representing uncertainty of accuracy values (which are ratios between two counts), the Beta distribution $\mathfrak{B}(\cdot, \cdot)$ is a natural choice. We call this baseline method **Beta-I**.

The variance of the estimated Beta density can be used for actively sampling arms. Ji et al. [7] describe a related scenario, stressing on active sampling. However, this approach cannot share observations or smooth the estimated density at a sparsely-populated arm with information from similar arms. In our real-life scenario, we expect accuracy surface smoother and the number of arms to be large enough that many arms will get very few, if any, instances.

The second baseline method, which we call **BernGP**, is to view the $(\boldsymbol{a}, c)$ instances in $D$ as a standard classification data set with the binary $c$ values as class label and $\boldsymbol{a}$ as input features. Given the limited data, we can use the well-known GP classification approach [9] for fitting smooth values $\rho$ as a function of $\boldsymbol{a}$. Suppose the arms $\boldsymbol{a}$ can be embedded to $\mathcal{V}(\boldsymbol{a})$ in a suitable space induced by some similarity kernel. In this embedding space, we expect the accuracy of $S$ to vary smoothly. Given a kernel $K_1(\boldsymbol{a}, \boldsymbol{a}')$ to guide the extent of sharing of information across arms, a standard form of this GP would be

$$P(c|\boldsymbol{a}) = \text{Bernoulli}(c; \text{sigmoid}(f_{\boldsymbol{a}})); \quad f \sim GP(0, K_1). \tag{3}$$

The GP can give estimates of uncertainty of $\rho(\boldsymbol{a})$, which may be used for active sampling of arms.

As we will demonstrate, such GP-imposed estimate of uncertainty of $\rho(\boldsymbol{a})$ is inadequate, because it loses sight of the number of supporting observations at each arm, which could be very diverse. This is because the standard GP assumption of homoscedasticity, that is, identical noise around each arm is violated when observations per arm differ significantly. We therefore need a mechanism to

separately account for the uncertainty at each arm, even the unexplored ones, to guide the strategy for actively collecting more labeled data.

## 3.1 The basic BetaGP proposal

We model arm-specific noise by allowing each arm to represent the uncertainty of $\rho_a$, not just by an underlying GP as in BernGP above, but also by a separate scale parameter. Further, the scale parameter is smoothed over neighboring arms using another GP. The influence of this scale on the uncertainty of $\rho_a$ is expressed by a Beta distribution as follows:

$$P(\rho|\boldsymbol{a}) \sim \mathfrak{B}(\rho; \phi(f_{\boldsymbol{a}}), \psi(g_{\boldsymbol{a}})) \tag{4}$$

$$\phi(f_{\boldsymbol{a}}) = \text{sigmoid}(f_{\boldsymbol{a}}), \qquad f \sim GP(0, K_1), \tag{5}$$

$$\psi(g_{\boldsymbol{a}}) = \log(1 + e^{g_a}), \qquad g \sim GP(0, K_2), \tag{6}$$

where we use $\phi(\bullet), \psi(\bullet)$ to denote the parameters of the Beta distribution at arm $\boldsymbol{a}$. The Beta distribution is commonly represented via $\alpha, \beta$ parameters whereas we chose the less popular mean ($\phi$) and scale ($\psi$) parameters. While these two forms are functionally equivalent with $\phi = \frac{\alpha}{\alpha+\beta}, \psi = \alpha + \beta$, we preferred the second form because imposing GP smoothness across arms on the mean accuracy and scale seemed more meaningful. We validate this empirically in the Appendix B.

Two kernel functions $K_1(\boldsymbol{a}, \boldsymbol{a}'), K_2(\boldsymbol{a}, \boldsymbol{a}')$ defined over pairs of arms $\boldsymbol{a}, \boldsymbol{a}' \in \mathcal{A}$ control the degree of smoothness among the Beta parameters across the arms. We use an RBF kernel defined over learned shared embeddings $\mathcal{V}(\boldsymbol{a})$:

$$K_1(\boldsymbol{a}, \boldsymbol{a}') = s_1 \exp\left[-\frac{\|\mathcal{V}(\boldsymbol{a})-\mathcal{V}(\boldsymbol{a}')\|^2}{l_1}\right], \qquad K_2(\boldsymbol{a}, \boldsymbol{a}') = s_2 \exp\left[-\frac{\|\mathcal{V}(\boldsymbol{a})-\mathcal{V}(\boldsymbol{a}')\|^2}{l_2}\right] \tag{7}$$

where $s_1, s_2, l_1, l_2$ denote the scale and length parameters of the two kernels. The scale and length parameters are learned along with the parameters of embeddings $\mathcal{V}(\boldsymbol{a})$ during training.

Initially, we assume we are given a labeled dataset $D = \{(\mathbf{x}_i, \boldsymbol{a}_i, y_i) : i = 1\ldots, I\}$ with attribute information available. Using predictions from the classification service $S$, we associate a 0/1 accuracy $c_i = \text{Agree}(y_i, S(\mathbf{x}_i))$. We can thus extend $D$ to $\{(\mathbf{x}_i, \boldsymbol{a}_i, y_i, c_i) : i \in [I]\}$.

Let $c_{\boldsymbol{a}} = \sum_{i:A(\mathbf{x}_i)=\boldsymbol{a}} c_i$ denote the total agree score in arm $\boldsymbol{a}$. Let $n_{\boldsymbol{a}}$ denote the total number of labeled examples in arm $\boldsymbol{a}$. The likelihood of all observations given functions $f, g$ decomposes as a product of Beta-binomial[3] distributions at each arm as follows:

$$\Pr(D|f, g) = \prod_{\boldsymbol{a}} \int_{\rho} \rho^{c_a}(1-\rho)^{n_{\boldsymbol{a}}-c_{\boldsymbol{a}}} \mathfrak{B}(\rho|\phi(f_{\boldsymbol{a}}), \psi(g_{\boldsymbol{a}})))\mathrm{d}\rho. \tag{8}$$

$$= \prod_{\boldsymbol{a}} \frac{\text{B}(\phi(f_{\boldsymbol{a}})\psi(g_{\boldsymbol{a}}) + c_a, (1 - \phi(f_{\boldsymbol{a}}))\psi(g_{\boldsymbol{a}}) + n_{\boldsymbol{a}} - c_a)}{\text{B}(\phi(f_{\boldsymbol{a}})\psi(g_{\boldsymbol{a}}), (1 - \phi(f_{\boldsymbol{a}}))\psi(g_{\boldsymbol{a}}))}, \tag{9}$$

where B is the Beta function, and the second expression is a rewrite of the Beta-binomial likelihood.

During training we calculate the posterior distribution of functions $f, g$ using the above data likelihood $\Pr(D|f, g)$ and GP priors given in eqns. (5) and (6). The posterior cannot be computed analytically given our likelihood, so we use variational methods. Further, we reduce the $\mathcal{O}(|\mathcal{A}|^3)$ complexity of posterior computation, using the inducing point method of Hensman et al. [9], whereby we learn $m$ locations $\mathbf{u} \in \mathbb{R}^{d \times m}$, mean $\mu \in \mathbb{R}^m$, and covariance $\Sigma \in \mathbb{R}^{m \times m}$ of inducing points. Doing so brings down the complexity to $\mathcal{O}(m^2|\mathcal{A}|), m \ll |\mathcal{A}|$. These parameters are learned end to end with the parameters of the neural network used to extract embeddings $\mathcal{V}(\boldsymbol{a})$ of arms $\boldsymbol{a}$, and kernel parameters $s_1, s_2, \ell_1, \ell_2$. We used off-the-shelf Gaussian process library: GPyTorch [10] to train the above likelihood with variational methods. Details of this procedure can be found in the Appendix C. We denote the posterior functions as $P(f|D), P(g|D)$. Thereafter, the mean estimated accuracy for an arm $\boldsymbol{a}$ is computed as

$$\mathbb{E}(\rho|\boldsymbol{a}) = \mathbb{E}_{f \sim P(f|D)}[\phi(f_{\boldsymbol{a}})]. \tag{10}$$

We call this setup **BetaGP**. Next, we will argue why BetaGP still has serious limitations, and offer mitigation measures.

---

[3]The $\binom{n_a}{c_a}$ term does not apply since we are given not just counts but accuracy $c_i$ of individual points.

## 3.2 Supervision for scale parameters

We had introduced the second GP $g_{\boldsymbol{a}}$ to model arm-specific noise, and similar techniques have been proposed earlier by Lázaro-Gredilla and Titsias [11], Kersting et al. [12], Goldberg et al. [13], but for heteroscedasticity in Gaussian observations. However, we found the posterior distribution of scale values $\psi(g_{\boldsymbol{a}})$ at each arm tended to converge to similar values, even across arms with orders of magnitude difference in number of observations $n_{\boldsymbol{a}}$. On hindsight, that was to be expected, because the data likelihood (8) increases monotonically with scale $\psi_{\boldsymbol{a}}$. The only control over its converging to $\infty$ is the GP prior $g \sim GP(0, K_2)$. In Appendix D, we illustrate this phenomenon with an example. We propose a simple fix to the scale supervision problem. We expect the relative values of scale across arms to reflect the distribution of the proportion of observations $\frac{n_{\boldsymbol{a}}}{n}$ across arms (with $n = \sum_{\boldsymbol{a}} n_{\boldsymbol{a}}$). We impose a joint Dirichlet distribution using the scale of arms $\psi(g_{\boldsymbol{a}})$ as parameters, and write the likelihood of the observed proportions as (with $\Gamma$ denoting Gamma function):

$$\log \Pr(\{n_{\boldsymbol{a}}\}|g) = \sum_{\boldsymbol{a}} ((\psi(g_{\boldsymbol{a}}) - 1) \log \frac{n_{\boldsymbol{a}}}{n} - \log \Gamma(\psi(g_{\boldsymbol{a}})) + \log \Gamma(\sum_{\boldsymbol{a}} \psi(g_{\boldsymbol{a}})) \qquad (11)$$

We call this **BetaGP-SL**. With this as an additional term in the data likelihood, we obtained significantly improved uncertainty estimates at each arm, as we will show in the experiment section.

## 3.3 Pooling for sparse observations

Recall that the observations are accumulation of 1/0 agreement scores for all instances that belong to an arm. Given the nature of our problem, arms have varying levels of supervision, and also highly varying true accuracy values. Even when the available labeled data is large, many arms will continue to have sparse supervision because they represent rare attribute combinations. The combination of high variance observations and sparse supervision could lead to learning of non-smooth kernel parameters. In Appendix D, we demonstrate with a simple setting that GP parameters learned on noisy observations under-represent the smoothness of the surface. The situation is further aggravated when learning a deep kernel. This problem has resemblance to "collapsing variance problem" [14] such as when Gaussian mixture models overfit on outliers or when topic models overfit a noisy document in the corpus. Instead of depending purely on GP priors to smooth over these noisy observations, we found it helpful to also externally smooth noisy observations. For each arm $\boldsymbol{a}$ with observations below a threshold, we mean-pool observations from some number of nearest neighbors, weighted by their kernel similarity with $\boldsymbol{a}$. We will see that such external smoothing resulted in significantly more accurate estimates particularly for arms with extreme accuracy values. We call this method **BetaGP-SLP** (note that this also includes the scale supervision objective described in the previous section). Two other mechanisms take us to the full form of the **AAA** system, which we describe next.

## 3.4 Exploration

The variance estimate of an arm informs its uncertainty and is commonly used for efficient exploration [15]. Let $P(f|D), P(g|D)$ denote the learned posterior distribution of the GPs. Using these, the estimated variance at an arm is given as:

$$\mathbb{V}(\rho|\boldsymbol{a}) = \mathbb{E}_{f \sim P(f|D), g \sim P(g|D)} \left[ \int_{\rho} (\rho - \mathbb{E}(\rho|\boldsymbol{a}))^2 \mathfrak{B}(\rho; \phi(f_{\boldsymbol{a}}), \psi(g_{\boldsymbol{a}})) d\rho \right] \qquad (12)$$

where the expected value is given in eqn. (10). We use sampling to estimate the above expectation. The arm to be sampled next is chosen as the one with the highest variance among unexplored arms. We then sample an unexplored example with highest affiliation ($P(\mathbf{a} \mid \mathbf{x})$) with the chosen arm.

## 3.5 Modeling Attribute Uncertainty

Recall that attributes of an instance $\mathbf{x}$ are obtained from models $M_k(a_k|\mathbf{x}), \ k \in [K]$, which may be highly noisy for some attributes. Thus, we cannot assume a fixed attribute vector $A(\mathbf{x})$ for an instance $\mathbf{x}$. We address this by designing a model that can combine these noisy estimates into a joint distribution $P(\mathbf{a}|\mathbf{x})$ using which, we can fractionally assign each instance $\mathbf{x}_i$ across arms. A baseline model for $P(\mathbf{a}|\mathbf{x})$ would be just the product $\prod_{k=1}^{K} M_k(a_k|\mathbf{x})$. However, we expect values of attributes to be correlated (e.g. attribute 'high-pitch' is likely to be correlated with gender 'female'). Also, the probabilities $M_k(a_k|\mathbf{x})$ may not be well-calibrated.

We therefore propose an alternative joint model that can both recalibrate individual classifiers via temperature scaling [16], and model their correlation. We have a small seed labeled dataset $D$ with gold attribute labels, independent noisy distributions from each attribute model $M_k(a_k|\mathbf{x})$, and an unlabeled dataset $U$. We prefer simple factorized models. We factorize $\log \Pr(\mathbf{a}|\mathbf{x})$ as a sum of temperature-weighted logits and a joint (log) potential as shown in expression (13) below.

$$\log \Pr(\mathbf{a}|\mathbf{x}) = \log \Pr(a_1, a_2, \cdots, a_K|\mathbf{x}) = \sum_{k=1}^{K} t_k \log M_k(a_k|\mathbf{x}) + N(a_1, a_2, \cdots, a_K) \quad (13)$$

Here $N$ denotes a dense network to model the correlation between attributes, and $t_1, \ldots, t_K$ denote the temperature parameters used to rescale noisy attribute probabilities. The maximum likelihood over $D$ is $\max_{t,N} \sum_{(\mathbf{x}_i, \mathbf{a}_i) \in D} \log \Pr(\mathbf{a}_i|\mathbf{x}_i)$

$$= \max_{t,N} \sum_{(\mathbf{x}_i, \mathbf{a}_i) \in D} \left\{ \sum_{k=1}^{K} t_k \log M_k(a_{ik}|\mathbf{x}_i) + N(a_{i1}, \ldots a_{iK}) - \log(Z_i) \right\} \quad (14)$$

$Z_i$ denotes the partition function for an example $\mathbf{x}_i$ which requires summation over $\mathcal{A}$. Exact computation of $Z_i$ could be intractable especially when $\mathcal{A}$ is large. In such cases, $Z_i$ can be approximated by sampling. In our case, we could get exact estimates.

In addition to $D$, we use the unlabeled instances $U$ with predictions from attribute predictors filling the role of gold-attributes. Details on how we train the parameters on large but noisy $U$ and small but correct $D$ can be found in the Appendix E.

The estimation method of BetaGP-SLP with variance based exploration and calibration described here constitute our proposed estimator: AAA. Detailed pseudo-code of AAA is given in the Appendix I.

## 4 Experiments

Our exploration of various methods and data sets is guided by the following research questions.

- How do various methods for arm accuracy estimation compare?
- To what extent do BetaGP, scale supervision and pooled observations help beyond BernGP?
- For the best techniques from above, how do various active exploration strategies compare?
- How well does our proposed model of attribute uncertainty work?

### 4.1 Data sets and tasks

We experiment with two real data sets and tasks. Our two tasks are male-female gender classification with two classes and animal classification with 10 classes.

**Male-Female classification (MF):** CelebA [17] is a popular celebrity faces and attribute data set that identifies the gender of celebrities among 39 other binary attributes. The label is gender. The accuracy surface spans various demographic, style, and personality related attributes. We hand-pick a subset of 12 attributes that include attributes that we deem important for gender classification among some other gender-neural attributes such as if the subject is young or wearing glasses (see Appendix F for more details). We used a random subset of 50,000 examples from the dataset for training classifiers on each of the 12 attributes using a pretrained ResNet-50 model. The remaining 150,000 examples in the data set are set as the unlabeled pool from which we actively explore new examples for human feedback. The twelve binary attributes make up for $2^{12} = 4,096$ attribute combinations.

**Animal classification (AC):** COCO-Stuff [18] provides an image collection. For each image, labels for foreground (cow, camel) and background (sky, snow, water) 'stuff' are available. Visual recognition models often correlate the background scene with the animal label such as camel with deserts and cow with meadows. Thus, foreground labels are our regular $y$-labels while background stuff labels supply our notion of attributes.

We collapse fine background labels into five coarse labels using the dataset provided label hierarchy. These are: water, ground, sky, structure, furniture. The Coco dataset has around 90 object (foreground)

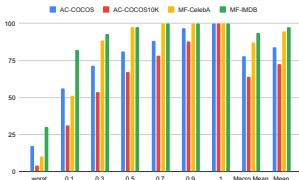

Figure 1: Macro and micro averaged accuracy (right most) and ten quantiles (x-axis) of per-arm accuracy (y-axis).

| Service→ | AC-COCOS10K | AC-COCOS | MF-IMDB | MF-CelebA |
|---|---|---|---|---|
| CPredictor | 5.4 / 15.0 | 3.2 / 9.4 | 1.2 / 8.2 | 5.2 / 35.9 |
| Beta-I | 7.0 / 15.6 | 4.3 / 10.0 | 1.6 / 8.4 | 4.7 / 30.3 |
| BernGP | 7.0 / 13.2 | 3.5 / 8.6 | 1.7 / 7.6 | 4.9 / 28.1 |
| BetaGP | 7.1 / 14.3 | 3.3 / 7.9 | 2.2 / 6.6 | 4.6 / 25.9 |
| BetaGP-SL | 5.3 / 11.7 | 2.8 / 6.8 | 1.4 / 4.4 | 4.1 / 22.6 |
| BetaGP-SLP | 4.7 / 10.4 | 2.8 / 5.7 | 1.4 / 3.9 | 4.3 / 23.3 |

Table 2: Comparing different estimation methods on labeled data size 2000 across four tasks. No exploration is involved. Each cell shows two numbers in the format "macro MSE / worst MSE" obtained over three runs. BetaGP-SLP generally gives the lowest MSE.

labels. Here we use a subset of 10 labels corresponding to animals. We take special care to filter out images with multiple/no animals and adapt the pixel segmentation/classification task to object classification (see the Appendix F for more details). The image is further annotated with the five binary labels corresponding to five coarse stuff labels. The scene descriptive five binary labels and ten object labels make up for $32 \times 10 = 320$ attribute combinations.

## 4.2 Service Models

For the MF task, we use two service models ($S$). **MF-CelebA** is a service model for gender classification. To simulate separate $D$ and $U$, it is trained on a random subset of CelebA with a ResNet50 model. **MF-IMDB** is a publicly available[4] classifier trained on IMBD-Wiki dataset, also using the ResNet50 architecture. The attribute predictors are trained using ResNet50 on a subset of the CelebA dataset for both service models.

For the AC task, we use two publicly available[5] service models ($S$). **AC-COCOS** was trained on COCOS data set with 164K examples. **AC-COCOS10k** was trained on COCOS10K, an earlier version of COCOS with only 10K instances. We use these architectures for both label and attribute prediction. See Appendix F, G for more details on accuracy surface, attribute predictor, service models and their architecture. In Figure 1, we illustrate some statistics of the shape of the accuracy surface for the four dataset-task combinations. Although $S$'s mean accuracy (rightmost bars) is reasonably high, the accuracy of the arms in the 10% quantile is abysmally low, while arms in the top quantiles have near perfect accuracy. This further motivates the need for an accuracy surface instead of single accuracy estimate.

## 4.3 Methods Compared

We compare the proposed estimation method AAA against natural baselines, alternatives, and ablations. Some of the methods, such as **Beta-I**, **BernGP** and **BetaGP**, we have already defined in Section 3. We train methods BernGP and BetaGP using the default arm-level likelihood. We also separately evaluate the impact of our fixes on BetaGP with only scale supervision: **BetaGP-SL** and along with mean pooling: **BetaGP-SLP**. We also include a trivial baseline: **CPredictor** which fits all the arms with a global accuracy estimated using gold $D$. We do not try sparse observation pooling with Beta-I since there is no notion of per-arm closeness. We also skip it on BernGP since it is worse than BetaGP as we will show below. Recall that Beta-I modeling is related to Ji et al. [7].

## 4.4 Other experimental settings

**Gold accuracies** $\rho(\boldsymbol{a})$: We compute the oracular accuracy per arm using the gold attribute/label values of examples in $U$ which we treat as unlabeled during exploration. For every arm with at least five examples, we set its accuracy to be the empirical estimate obtained through the average correctness of all the examples that belong to the arm. We discard and not evaluate on any arms with fewer than five examples since their true accuracy cannot reliably be estimated.

**Warm start:** We start with 500 examples having gold attributes+labels to warm start all our experiments. The random seed also picks this random subset of 500 labeled examples. We calculate the

---

[4]https://github.com/yu4u/age-gender-estimation
[5]https://github.com/kazuto1011/deeplab-pytorch/

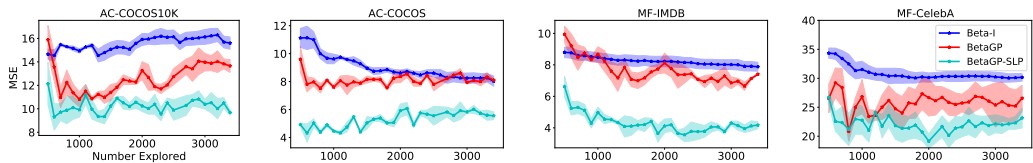

Figure 3: Comparison of estimation methods using worst MSE metric. The shaded region shows standard error. BetaGP-SLP consistently performs better than BetaGP. Beta-I is worse than its smoother counterparts.

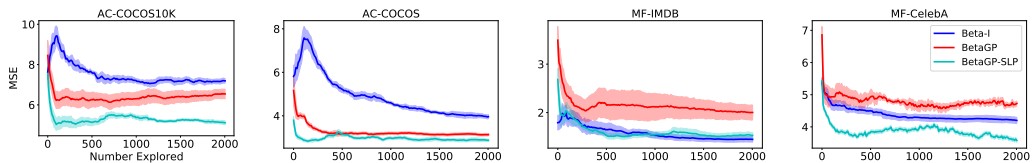

Figure 4: Comparison of exploration methods. BetaGP-SLP reduces macro MSE fastest most of the time. Shaded region shows standard error.

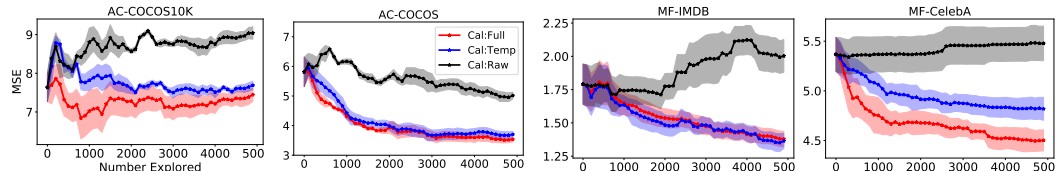

Figure 5: Calibration methods compared on different tasks. Cal:Full (red) includes temperature-based recalibration and correlation modeling with joint potential and gives the best macro MSE. Shaded region shows standard error.

overall accuracy of the classifier on these warm start examples as $\hat{\rho} = (\sum_i c_i)/(\sum_i 1)$. For all arms we warm start their observation with $c_{\boldsymbol{a}} = \lambda\hat{\rho}, n_{\boldsymbol{a}} = \lambda$ where $\lambda = 0.1$, a randomly picked low value.

Unless otherwise specified, we give equal importance to each arm and report MSE macroaveraged over all arms. Along with macro MSE, we also sometimes report MSE on the subset of 50 worst (true-)accuracy arms, referred to as worst MSE. We report other aggregate errors in the Appendix H. All the numbers reported here are averaged over three runs each with different random seed. The initial set of warm-start examples ($D$) is also changed between the runs. In the case of BetaGP-SLP, for any arm with observation count below 5, we mean pool from its three closest neighbours.

In the following Sections: 4.5 and 4.6, we compare various estimation and exploration strategies with $P(\mathbf{a}|\mathbf{x})$ noise calibrated as described in Section 3.5. In Section 4.7, we study different forms of calibration and demonstrate the superiority of our proposed calibration technique of Equation (13).

## 4.5 Accuracy Estimation Quality

We evaluate methods on their estimation quality when each method is provided with exactly the same (randomly chosen) labeled set. We compare the four service models when fitted on labeled data of size 2,000 and the results appear in Table 2. Note that we only have label supervision on $\mathcal{Y}$ in the labeled data. Table 2 shows macro and worst MSE, standard deviation for each metric can be found in Appendix H. In Figure 3, we show worst MSE for a range of labeled data sizes along with their error bars. We make the following observations. **Smoothing helps:** Since we have a large number of arms, we expect Beta-I to fare worse than its smooth counterparts (BernGP and BetaGP), especially on the worst arms. This is confirmed in the table. In three out of four cases, Beta-I method is worse than even the constant predictor CPredictor on both metrics. **Modeling arm specific noise helps:** BetaGP is better than BernGP on almost all the cases in the table. **Significant gains when the scale supervision problem of BetaGP is fixed:** BetaGP-SL is significantly better than BetaGP in the table and figure. **Our pooling strategy helps:** BetaGP-SLP improves BetaGP-SL over worst MSE without hurting macro MSE as seen in the table and figure.

### 4.6 Exploration Efficiency

We compare different methods that use their own estimated variance for exploring instances to label (Section 3.4), as a function of the number of explored examples — see Figure 4. In most cases, BetaGP-SLP gives the smallest macro MSE, beating Beta-I and BetaGP. Note Beta-I is the exploration method recently suggested in [7]. We observe that BetaGP provides very poor exploration quality, indicating that the uncertainty of arms is not captured well by just using two GPs. In fact, in many cases BetaGP is worse than Beta-I, even though we saw the opposite trend in estimation quality (Figure 3). These experiments brings out the significant role of Dirichlet scale supervision and pooled observations in enhancing the uncertainty estimates at each arm.

### 4.7 Impact of Calibration

We consider two baselines along with our method explained in Section 3.5: **Cal:Raw**, which uses the predicted attribute from the attribute models without any calibration and **Cal:Temp**, which calibrates only the temperature parameters shown in eqn. (13), i.e., without the joint potential part. We refer to our method of calibration using temperature and joint potentials as **Cal:Full**. We compare these on the four tasks with estimation method set to Beta-I and random exploration strategy. Figure 5 compares the three methods: Cal:Raw(Black), Cal:Temp(Blue), Cal:Full(Red). The X-axis is the number of explored examples beyond $D$, and Y-axis is estimation error. Observe how Cal:Temp and Cal:Full are consistently better than Cal:Raw, and Cal:Full is better than Cal:Temp.

## 5 Related Work

Our problem of actively estimating the accuracy *surface* of a classifier generalizes the more established problem of estimating a single accuracy *score* [19, 20, 21, 22, 23, 24]. For that problem, a known solution is stratified sampling, which partitions data into homogeneous strata and then seeks examples from regions with highest uncertainty and support. If we view each arm as a stratum, our method follows similar strategy. A key difference in our setting is that low support arms cannot be ignored. This makes it imperative to calibrate well the uncertainty under limited and skewed support distribution. The setting of Ji et al. [7] is the closest to ours. However, their work only considers a single attribute which they fit using Beta-I, whereas we focus on the challenges of estimating accuracy over many sparsely populated attribute combinations.

**Sub-population performance:** Several recent papers have focused on identifying sub-populations with significantly worse accuracy than aggregated accuracy [2, 3, 6, 8, 25, 26]. Some of these have also proposed sample-efficient techniques [6, 8] for estimation of performance on specific sub-groups, such as the ones defined by attributes like gender and race. Our accuracy surface estimation problem can be seen as a generalization where we need to estimate for all sub-groups defined in the Cartesian space of pre-specified semantic attributes. Mitchell et al. [5] recommend reporting model performance under the influence of various relevant demographic/environmental factors as model cards–similar to the accuracy surface.

**Experiment design:** Another related area is experiment design using active explorations with GPs [27]. Their goal is to find the mode of the surface whereas our goal is to estimate the entire surface. Further, each arm in our setting corresponds to multiple instances, which gives rise to a degree of heteroscedasticity and input-dependent noise that is not modeled in their settings. Lázaro-Gredilla and Titsias [11], Kersting et al. [12] propose to handle heteroscedasticity by using a separate GP to model the variance at each arm. However, we showed the importance of additional terms in our likelihood and observation pooling to reliably represent estimation uncertainty. Wenger et al. [28] propose observation pooling for estimating smooth Betas but they assume a fixed kernel.

**Model debugging:** Testing deep neural network (DNN) is another related emerging area [29]. Pei et al. [30], Tian et al. [31], Sun et al. [32], Odena et al. [33] propose to generate test examples with good coverage over all activations of a DNN. Ribeiro et al. [34], Kim et al. [35] identify rules that explain the model predictions.

## 6 Conclusion

We presented AAA, a new approach to estimate the accuracy of a classification service, not as a single number, but as a surface over a space of attributes (arms). AAA models uncertainty with a

Beta distribution at each arm and regresses these parameters using two Gaussian Processes to capture smoothness and generalize to unseen arms. We proposed an additional Dirichlet likelihood to mitigate an over-smoothing problem with GP's estimation of Beta distributions' scale parameters. Further, to protect these high-capacity GPs from unreliable accuracy observations at sparsely populated arms, we propose to use an observation pooling strategy. Finally, we show how to handle noisy attribute labels by an efficient joint recalibration method. Evaluation on real-life datasets and classification services show the efficacy of AAA, both in estimation and exploration quality.

**Limitation and future work:** (1) We have evaluated AAA on the order of thousands of arms. Even larger attribute spaces could unearth more challenges. (2) Identifying relevant attributes for an application can be non-trivial. Future work could devise strategies for attribute selection. (3) Characterizing test-time data shifts could in itself be hard, particularly for text — there could be subtle changes in word usage, style, or punctuation. A more expressive attribute space needs to be developed for text applications.

# 7   Acknowledgements

The first author is supported by Google PhD Fellowship. This research was partly sponsored by the IBM AI Horizon Networks - IIT Bombay initiative.

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
