# Active Assessment of Prediction Services as Accuracy Surface Over Attribute Combinations
# (Appendix)

| Main Section | Appendix |
|---|---|
| Source Code | Appendix A |
| Section 3.1 | Appendix B, C |
| Section 3.2 | Appendix D |
| Section 3.5 | Appendix E, I |
| Section 4.1, 4.2 | Appendix F, G |
| Section 4.4, 4.5 | Appendix H |

Table 6: Mapping between main and appendix sections.

## A  Source Code

Our code, dataset and instructions for replicating the results can be found at this link.

## B  Parametric Form of BetaGP

In Section 3.1, we claimed that BetaGP with (mean, scale) parameterization is better than BetaGP$\alpha\beta$ with the standard $(\alpha, \beta)$ parameterization of the Beta distribution. In this section, we present some empirical evidence corroborating the claim.

We compare between the two parametric forms with two service models in Table 7. For BetaGP$\alpha\beta$, we use two GPs, one to approximate the latent value corresponding to $\alpha$, and other for $\beta$. We use soft-plus operation to transform the latent values to their admissible positive $\alpha, \beta$ values.

We report macro-averaged mean square errors on two tasks in Table 7, when fitting on 2,000 instances — similar to the setting of Section 4.5. We found the BetaGP$\alpha\beta$ estimates unstable and far worse, perhaps because smoothness is not expected in either of $\alpha, \beta$ parameters across arms making the GP's bias ineffective.

| Service→ | MF-CelebA | | AC-COCOS | |
|---|---|---|---|---|
| Method→ | BetaGP | BetaGP$\alpha\beta$ | BetaGP | BetaGP$\alpha\beta$ |
| 1000 | 5.4 / 0.5 | 6.6 / 0.1 | 3.7 / 0.2 | 5.6 / 0.6 |
| 2000 | 4.6 / 0.8 | 6.2 / 0.1 | 3.3 / 0.2 | 4.8 / 0.1 |
| 3500 | 4.6 / 0.3 | 6.1 / 0.1 | 3.2 / 0.2 | 5.2 / 0.4 |

Table 7: Comparison of estimation error between BetaGP with (mean, scale) parameterization vs. BetaGP$\alpha\beta$. BetaGP$\alpha\beta$ is worse than BetaGP.

## C  More Details of Gaussian Process (GP) Setup

In this section, we give further details on GP training, posterior approximation and computational cost. This section elaborates on Section 3.1.

In all our proposed estimators, the data likelihood is modeled either by a Bernoulli or a Beta distribution. Data likelihood term of BernGP, BetaGP, are shown in Eqn. (3), Eqn. (9), respectively. Due to the non-Gaussian nature of the data likelihood, the posterior on parameters cannot be expressed in a closed form. Several approximations exist for fitting the posterior especially for the more standard BernGP, we will discuss one such method in what follows. Recall that we model two latent values $f, g$ each modeled by an independent GP. They can be seen to have been drawn from a single GP with even larger dimension and with appropriately defined kernel matrix. For the sake of explanation and with a slight abuse of notation, we denote by $\mathbf{f}$, the concatenation of $f$ and $g$. The corresponding

kernel for the concatenated vector is appropriately made by combining the kernels of either of the latent values with kernel entries corresponding to interaction between $f$ and $g$ set to 0.

Variational methods are popular for dealing with non-Gaussian likelihoods in GP. In this method, we fit a multi-variate Gaussian that closely approximates the posterior, i.e. minimizes $\mathcal{D}_{\mathrm{KL}}(q(\mathbf{f})\|P(\mathbf{f}|D))$. GPs are often used in their sparse avatars using *inducing points* [36, 37] that provide approximations to the full covariance matrix with large computation benefits. As a result, q(f) is parameterized by the following trainable parameters (let 'd', 'm' denote the input dimension and number of inducing points resp.): (a) Z, a matrix of size $d \times m$, of locations of 'm' inducing points (b) $\mu \in \mathbb{R}^m, \Sigma \in \mathbb{R}^{m \times m}$, denoting mean and covariance of the inducing points. In order to minimize $\mathcal{D}_{\mathrm{KL}}(q(\mathbf{f})\|P(\mathbf{f}|D))$, a pseudo objective called Evidence Lower Bound (ELBO), shown below, is employed:

$$q^*(\mathbf{f}) = \underset{\mathbf{f} \sim q(\mathbf{f})}{\operatorname{argmax}} \mathbb{E}_q[\log P(D|\mathbf{f})] - \mathcal{D}_{\mathrm{KL}}(q(\mathbf{f})\|P(\mathbf{f})) \tag{15}$$

The first term above in Eqn. (15) maximizes data likelihood, which is Equation 9 in our case. The second term is a regularizer that regresses the posterior fit $q(\mathbf{f})$ close to the prior distribution $P(\mathbf{f})$ which is set to standard Normal. We optimize using this objective over all the parameters involved through gradient descent. The required integrals in (15) can be computed using Monte Carlo methods [9]. We describe further implementation details in the next section.

### Implementation Details

We use routines from GPytorch[6] library to implement the variational objective. Specifically, we extend ApproximateGP with VariationalStrategy, both of which are GPytorch classes, and set them to learn inducing point locations.

Number of *inducing points* when set to a very low value could overly smooth the surface and can have high computation overhead when set to a large value. We set the number of inducing points to 50 for all the tasks. The choice of 50 over a larger number is only to ensure reasonable computation speed.

Since we keep getting more observations as we explore, we use the following strategy for scheduling the parameter updates. We start with the examples in the seed set $D$ and update for 1,000 steps. We explore using the variance of the estimated posterior. We pick 12 arms with highest variance and label one example for each arm. After every new batch of observations, we make 50 update steps on all the data. As a result, we keep on updating the parameters as we explore more. We use Adam Optimizer with learning rate $10^{-3}$. At each step, we update over observations from all the arms. The flow of the exploration and parameter update is also shown in Algorithm 1.

We use the feature representations of the network used to model joint potentials described in Section 3.5 to also initialize the deep kernel induced by $\mathcal{V}$. A final new linear layer of default output size 20 is added to project the feature representations.

In our proposed method BetaGP-SLP, described in Section 3.3, we take the kernel average of three neighbours for any arm with fewer than five observations.

## D   Simple Setting

In Section 3.2, we describe how the objective of BetaGP does not supervise the scale parameter. Further, in Section 3.3, we posit that the presence of sparse observations leads to learning a non-smooth kernel. In this section, we illustrate these two observations using a simple setting.

We consider a simple estimation problem with 10 arms, their true accuracies go from 0.1 to a large value of 0.9 and then back to a small value of 0.2 as shown in the Table 8. In Table 8, we also show the number of observations per arm; Observe that the first three and the last three arms are sparsely observed.

We now present the fitted values by some of the methods we discussed in the main section. The index of an arm is the input for any estimator with no feature learning. Our motivation for discussing the simple setting is to illustrate the two limitations we discussed in the main content regarding the BetaGP objective: (a) the scale parameter of the BetaGP objective is not supervised (b) sparse observations lead to non-smooth surface. Toward these ends, we evaluate BetaGP, BetaGP-SL, BetaGP-SLP

---

[6]https://gpytorch.ai/

| Arm Index | 1 | 2 | 3 | 4 | 5 | 6 | 7 | 8 | 9 | 10 |
|---|---|---|---|---|---|---|---|---|---|---|
| Accuracy | 0.1 | 0.3 | 0.5 | 0.7 | 0.8 | 0.9 | 0.6 | 0.4 | 0.3 | 0.2 |
| N | 1 | 1 | 1 | 20 | 20 | 20 | 20 | 1 | 1 | 1 |
| Estimated Scale Value | | | | | | | | | | |
| BetaGP | 10.33 | 10.88 | 11.37 | 11.73 | 11.92 | 11.91 | 11.72 | 11.34 | 10.85 | 10.29 |
| BetaGP-SL | 1.49 | 1.42 | 1.59 | 9.64 | 10.40 | 10.42 | 9.58 | 1.59 | 1.42 | 1.49 |
| BetaGP-SLP | 1.72 | 1.63 | 1.92 | 9.63 | 10.38 | 10.27 | 9.48 | 1.93 | 1.53 | 1.62 |

Table 8: Arms, their indices, accuracies and number of observations (N) in the simple setting are shown in first three columns in that order. The scale parameter estimated using one of the algorithms for each arm is shown in the last three columns. Notice that BetaGP fitted scale parameter does not reflect the underlying observation sparsity for the first and last three arms.

methods on this setting. The fitted scale parameters for each arm by each of the estimators is shown in Table 8. Observe that BetaGP fitted scale parameter does not reflect the underlying observation sparsity of the first, last three arms. BetaGP-SL, BetaGP-SLP fitted scale values more faithfully reflect the underlying number of observations. All the numbers reported here are averaged over 20 seed runs.

| Method | Bias$^2$ | Variance | MSE |
|---|---|---|---|
| BetaGP | 0.052 | 1.011 | 1.063 |
| BetaGP-SL | 0.051 | 1.010 | 1.061 |
| BetaGP-SLP | 0.095 | 0.221 | **0.316** |

Table 9: Bias-variance decomposition of MSE in the simple setting

In Table 9, we show the bias$^2$, variance decomposition of the mean squared error from 20 independent runs. Observe that BetaGP, BetaGP-SL have low bias but large variance and BetaGP-SLP has much lower variance at a slight expense of bias, as a result the overall MSE value for BetaGP-SLP is much lower than the other two. Moreover, we look at the fitted kernel length parameter (recall from Equation (7)) as a proxy for smoothness of the fitted kernel. Large kernel length is indicative of long range smoothness. The average kernel length for BetaGP, BetaGP-SL, BetaGP-SLP are 0.67, 0.68, 1.87 respectively. Despite using a GP kernel, we find the estimates of BetaGP, BetaGP-SL of high variance, that is also indicative of short range smoothness apparent from the low average kernel length. On the other hand, BetaGP-SLP imposes long range smoothness, as a result, decreases the MSE value more effectively when compared with BetaGP-SL.

## E  Calibration Training Details

In this section, we give further training details on the noise calibration method discussed in Section 3.5.

As discussed in Section 3.5, we use both labeled, small $D$ and large $U$ for training calibration parameters that are expressed in the objective (14). On the unlabeled data $U$, we use attribute values predicted using the predictors: $\{M_k \mid k \in A\}$ as a proxy for true values. The use of predicted values as the replacement for true value under-represents the attribute prediction error rate and interferes in the estimation of temperature parameters $t$. However, if we use $U$ for training, we see a lot more attribute combinations and this can help identify more natural attribute combinations aiding in the learning of joint potential parameters of $N$. We mitigate the temperature estimation problem by up-sampling instances in $D$ such that the loss in every batch contains equal contribution from $D$ and $U$.

Recall that the MLE objective (14), contains contribution from two terms: (a) temperature scaled logits (b) attribute combination potential. In practice, we found that the second term (b) dominates the first, this causes under-training of the temperature parameters. Ideally, the two terms should be comparable and replaceable. We address this issue by dropping the second term corresponding to the network-assigned edge potential term in the objective half the times, which estimates better the temperature parameters. Further, we use a small held out fraction of $D$ for network architecture search on $L$, and for early stopping. The training procedure is summarized in Alg. 2.

# F More Task and Dataset Details

## F.1 MF-CelebA, MF-IMDB

As mentioned in Section 4.1, we hand-picked 12 binary attributes relevant for gender classification of the 40 total available attributes in the CelebA dataset. The twelve binary attributes are listed in Table 10, these constitute the $A$. $\mathcal{A}$ is the combination of twelve binary attributes and is $2^{12} = 4,096$ large. The attributes related to hair color are retained in this list due to the recent finding that hair-color is spuriously correlated with the gender in CelebA [3]. We ignored several other gender-neutral or rare attributes.

| Index | Name | Num. labels |
|---|---|---|
| 1 | Black Hair? | 2 |
| 2 | Blond Hair? | 2 |
| 3 | Brown Hair? | 2 |
| 4 | Smiling? | 2 |
| 5 | Male? | 2 |
| 6 | Chubby? | 2 |
| 7 | Mustache? | 2 |
| 8 | No Beard? | 2 |
| 9 | Wearing Hat? | 2 |
| 10 | Blurry? | 2 |
| 11 | Young? | 2 |
| 12 | Eyeglasses? | 2 |

Table 10: Attribute list of MF-CelebA, MF-IMDB.

## F.2 AC-COCOS, AC-COCOS10K

COCOS is a scene classification dataset, where pixel level supervision is provided. Methods are usually evaluated on pixel level classification accuracy. For simplicity, we cast it in to an object recognition task. The subset of ten animal labels we consider is shown in Table 11. We consider in our task five coarse background (stuff) labels by collapsing the fine labels to coarse using the label hierarchy shown in Table 12 and is the same as the official hierarchy [18].

We now describe how we cast the scene label classification task to an animal classification task. We first identify the subset of images in the train and validation set of the COCOS dataset which contain only one animal label, it could contain multiple background labels. If the image contains multiple animals, we exclude it, leaving around 23,000 images in the dataset. This is implemented in the routine: `filter_ids_with_single_object` of `cocos3.py` in the attached code. We also retrofit the scene classification models for animal classification. When the model (service model) labels pixels with more than one animal label, we retain the label associated with the largest number of pixels. This is implemented in `fetch_preds` routine of `cocos3.py`. Recall our calibration method makes use of the logit scores given by the attribute predictors, since we are aggregating prediction from multiple pixels, we do not have access to the logit scores. We simply set the logit score to +1 if a label is found in the image and -1 otherwise.

We follow the same protocol for both the tasks: AC-COCOS, AC-COCOS10K. The only difference between the two is the service model, AC-COCOS is a stronger service model trained on 164K size training data compared to AC-COCOS10K which is a model trained on a previous version of the dataset that is only 10K large. In these tasks, we use the same model for predicting the attributes and task labels since the pre-trained model we use is a scene label classifier. Both the pretrained[7] models were trained using ResNet101 architecture.

---

[7] https://github.com/kazuto1011/deeplab-pytorch

| Name |
|---|
| bird |
| cat |
| dog |
| horse |
| sheep |
| cow |
| elephant |
| bear |
| zebra |
| giraffe |

Table 11: List of ten animals in AC tasks

| Coarse label | Example of constituent stuff classes |
|---|---|
| water-other | sea, river |
| ground-other | ground-other, playingfield, platform, railroad, pavement |
| sky-other | sky-other, clouds |
| structural-other | structural-other, cage, fence, railing, net |
| furniture-other | furniture-other, stairs, light, counter, mirror-stuff |

Table 12: 80 stuff labels in the COCOS dataset are collapsed in to five coarse labels. Few examples are shown for each coarse label in the right column.

# G  Statistics of Accuracy Surface

We show in Table 13, details about our two data sets such as the number of attributes, number of arms and number of active arms. Active arms are arms with a support of at least five and are the ones used for evaluation. Large number of arms as shown in the table exclude the possibility of manual supervision, since it is hard to obtain and label data that covers all the arms.

In Figures 14a and 14b, we show some statistics that illustrate the shape of the accuracy surface. We note that, although the service model's mean accuracy is high, the accuracy of the arms in the 10% quantile is abysmally low while arms in the top-quantiles have near perfect accuracy. This further motivates for why we need an accuracy surface instead of single accuracy estimate.

| Dataset | # attributes | # arms | # active arms |
|---|---|---|---|
| CelebA | 12 | 4096 | 398 |
| COCOS | 6 | 320 | 176 |

Table 13: Attribute statistics per dataset. First and second column show number of attributes and total possible combinations of the attributes. Third column shows number of attribute combinations (arms) with at least a support of five in the unlabeled data. These are the arms on which accuracy surface is evaluated.

# H  Standard Deviation and More Evaluation Metrics

We include standard deviation accompanying numbers in Table 2 for macro MSE in Table 15 and for worst MSE in Table 17.

In the main content of the paper, we gave results using macro and worst MSE. In this following section, we show results using two other metrics. We follow the same setup as in Section 4.5.

**Micro-averaged MSE:** We assign importance to each arm based on its support. The error per arm is multiplied by its support (in $U$). Results with this error are shown in Table 16. The best predictor with this metric is the point estimate given by the CPredictor estimator which is not surprising since very few arms with high frequency dominate this metric.

**Infrequent MSE:** In Table 18, we show MSE evaluated only on the 50 arms that are least frequent in $U$.

For each of the above evaluation metrics, the trend between BetaGP BetaGP-SL, BetaGP-SLP is statistically significant.

# I  Pseudocode

The full flow of estimation and exploration is summarized in Algorithm 1. Algorithm 2 shows the calibration training sub-routine.

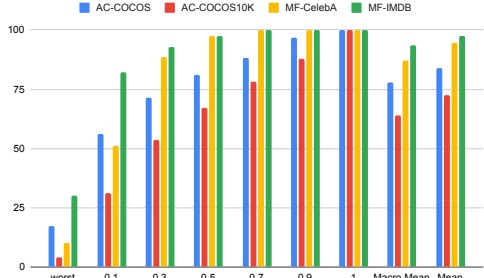

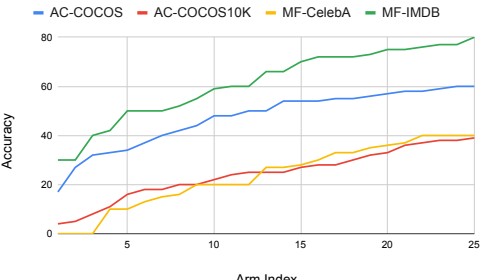

(a) We show the mean and ten quantiles of per-arm accuracy: 0, 0.1, 0.3, 0.5, 0.7, 0.9, 1. for each task when evaluated on their corresponding dataset (quantile 0 corresponds to the worst value). Observe the disparity between the best and the worst arms in terms of accuracy. In all the cases, also note how the large mean accuracy (macro-averaged over arms) does not do justice to explaining the service model's vulnerabilities.

(b) We show here the 25 worst arm accuracies for each of the service models. The large number of arms with accuracies much worse than mean accuracy further illustrates our argument for why we need accuracy surfaces.

Figure 14: Arm accuracies.

| Service ↓ | CPredictor | Beta-I | BernGP | BetaGP | BetaGP-SL | BetaGP-SLP |
|---|---|---|---|---|---|---|
| AC-COCOS10K | 5.4 / 0.2 | 7.0 / 0.6 | 7.0 / 0.7 | 7.1 / 0.3 | 5.3 / 0.2 | 4.7 / 0.1 |
| AC-COCOS | 3.2 / 0.1 | 4.3 / 0.3 | 3.5 / 0.3 | 3.3 / 0.2 | 2.8 / 0.0 | 2.8 / 0.1 |
| MF-IMDB | 1.2 / 0.0 | 1.6 / 0.1 | 1.7 / 0.2 | 2.2 / 0.2 | 1.4 / 0.1 | 1.4 / 0.1 |
| MS-CelebA | 5.2 / 0.1 | 4.7 / 0.2 | 4.9 / 0.4 | 4.6 / 0.8 | 4.1 / 0.1 | 4.3 / 0.1 |

Table 15: *Macro-averaged* MSE along with standard deviation on all tasks. Shown after trailing '/' is the standard deviation.

| Service ↓ | CPredictor | Beta-I | BernGP | BetaGP | BetaGP-SL | BetaGP-SLP |
|---|---|---|---|---|---|---|
| AC-COCOS10K | 3.0 / 0.0 | 3.4 / 0.2 | 3.6 / 0.2 | 3.5 / 0.1 | 3.2 / 0.1 | 3.3 / 0.4 |
| AC-COCOS | 1.4 / 0.0 | 1.8 / 0.2 | 1.6 / 0.1 | 1.6 / 0.1 | 1.7 / 0.1 | 2.1 / 0.3 |
| MF-IMDB | 0.2 / 0.0 | 0.3 / 0.1 | 0.2 / 0.0 | 0.3 / 0.0 | 0.3 / 0.1 | 0.8 / 0.1 |
| MF-CelebA | 0.8 / 0.0 | 0.7 / 0.1 | 0.7 / 0.1 | 0.7 / 0.2 | 0.9 / 0.1 | 1.2 / 0.1 |

Table 16: *Micro-averaged* MSE along with standard deviation on all tasks. Shown after trailing '/' is the standard deviation.

| Service ↓ | CPredictor | Beta-I | BernGP | BetaGP | BetaGP-SL | BetaGP-SLP |
|---|---|---|---|---|---|---|
| AC-COCOS10K | 15.0 / 0.8 | 15.6 / 0.3 | 13.2 / 2.2 | 14.3 / 3.0 | 11.7 / 1.7 | 10.4 / 1.5 |
| AC-COCOS | 9.4 / 0.4 | 10.0 / 0.4 | 8.6 / 0.7 | 7.9 / 0.9 | 6.8 / 0.5 | 5.7 / 0.5 |
| MF-CelebA | 8.2 / 0.2 | 8.4 / 0.7 | 7.6 / 1.3 | 6.6 / 0.7 | 4.4 / 0.6 | 3.9 / 0.7 |
| MF-IMDB | 35.9 / 0.6 | 30.3 / 1.2 | 28.1 / 2.7 | 25.9 / 2.7 | 22.6 / 1.4 | 23.3 / 2.3 |

Table 17: *Worst* MSE along with standard deviation on all tasks. Shown after trailing '/' is the standard deviation.

| Service ↓ | CPredictor | Beta-I | BernGP | BetaGP | BetaGP-SL | BetaGP-SLP |
|---|---|---|---|---|---|---|
| AC-COCOS10K | 7.3 / 0.3 | 11.8 / 1.5 | 10.8 / 1.9 | 12.4 / 0.1 | 7.3 / 0.2 | 6.4 / 0.4 |
| AC-COCOS | 4.0 / 0.1 | 6.9 / 1.2 | 4.8 / 0.3 | 4.9 / 0.3 | 3.8 / 0.1 | 3.8 / 0.3 |
| MF-CelebA | 2.9 / 0.0 | 2.8 / 0.0 | 3.3 / 1.2 | 3.9 / 0.4 | 3.2 / 0.2 | 2.9 / 0.3 |
| MF-IMDB | 11.7 / 0.2 | 11.3 / 0.3 | 11.4 / 0.6 | 11.0 / 0.7 | 9.3 / 1.1 | 9.4 / 0.6 |

Table 18: *Infrequent* MSE along with standard deviation on all tasks. Shown after trailing '/' is the standard deviation.

---

**Algorithm 1** AAA: Accuracy Surface Estimator and Active Sampler

---

**Require:** $D, U, \{M_k, k \in A\}, S, A, \mathcal{A}, \lambda, b$          ▷ Strength of prior ($\lambda$), budget (b)
1: $t^*, N^* \leftarrow \text{Calibrate}(\{M_k\}, D, U)$          ▷ Calibration training of Alg: 2
2: $P(\boldsymbol{a} \mid \mathbf{x}; t^*, N^*)$ is defined in Equation (13)
3: $\kappa = \frac{1}{|D|} \sum_{(x,y,a) \in D} \text{Agree}(S(x), y)$          ▷ Prior accuracy
4: c0, c1 $= \lambda \mathbb{1}_{|\mathcal{A}|}(1 - \kappa), \lambda \mathbb{1}_{|\mathcal{A}|}\kappa$          ▷ Initialize observation accumulators
5: E $= \emptyset$          ▷ Set of explored examples
6: **for** $(\mathbf{x}, y, \mathbf{a}) \in D$ **do**          ▷ Warm start with $D$
7:      $c0[\mathbf{a}] = c0[\mathbf{a}] + (1 - \text{Agree}(S(\mathbf{x}), y))$
8:      $c1[\mathbf{a}] = c1[\mathbf{a}] + \text{Agree}(S(x), y)$
9: **end for**
10: Initialize $\rho(\mathbf{a})$ with two GPs as described in Section 3.1, Equation (4)
11: Fit $\rho(\mathbf{a})$ on c0, c1 as shown in (9), (15).
12: **while** $|E| < b$: **do**
13:      $\hat{\boldsymbol{a}} = \text{argmax}_{\boldsymbol{a} \in \mathcal{A}} \mathbb{V}[\rho(\boldsymbol{a})]$          ▷ Pick the arm with highest variance
14:      $\hat{\mathbf{x}} = \text{argmax}_{\{x \in U, x \notin E\}} P(\hat{\boldsymbol{a}} \mid \mathbf{x})$          ▷ Unexplored arm with highest affiliation
15:      Add $\hat{x}$ to E
16:      Obtain the true label of $\hat{x}$: $\hat{y}$
17:      c = $\text{Agree}(S(\hat{x}), \hat{y})$
18:      **for** $a \in \mathcal{A}$ **do**
19:          c0[$a$] = c0[$a$] + (1-c)P($\hat{a}$ | $\hat{x}$)
20:          c1[$a$] = c1[$a$] + cP($\hat{a}$ | $\mathbf{x}$)
21:      **end for**
22:      Fit again as in Line 11
23: **end while**
24: **return** $\rho$

---

---

**Algorithm 2** Attribute Model Calibration (Section 3.5)

---

**Require:** $D, U, \{M_k, k \in A\}, \eta$
1: Initialize $t, N$
2: converged = False, $\tau = 10^{-3}$
3: **while** not converged **do**
4:      d, u$'$ $\leftarrow$ batch(D), batch(U)          ▷ sample a subset for batch processing
5:      u = $\{(\mathbf{x}, \{M_k(\mathbf{x}) \mid k \in A\})$ for $\mathbf{x}$ in u$'\}$
6:      LL = Eqn (14) on d, u
7:      t, N = optimizer-update($\eta, \nabla_t LL, \nabla_N LL$)
8:      converged = True if LL < $\tau$
9: **end while**
10: **return** $t, N$

---