# OpenReview forum: "Active Assessment of Prediction Services as Accuracy Surface Over Attribute Combinations"
_NeurIPS.cc/2021/Conference — NeurIPS 2021 Poster_

### Official Review · Reviewer_U6pf · 2021-07-13

**Rating:** 6
**Confidence:** 4

**Summary:**

        This paper describes a method to estimate the predictive performance of a black-box classifier on different tasks depending on the subsets of attributes considered. Some level of smoothness is assumed depending on the attributes considered for prediction. This smoothness is captured using a Gaussian process. The model also uses a beta distribution as the generative model for the probability of correctly predicting a label for a data instance. Further improvements are introduced in the model using a Dirichlet distribution. The proposed method is evaluated on several tasks and compare with several base-lines that are variants of the proposed method. An active learning strategy that queries the points with the highest associated variance is suggested.


**Limitations And Societal Impact:**

Yes, about the limtiations. The authors do not forsee any negative impact.

**Main Review:**

I believe that this is a well written paper that proposes an interesting model to address the problem described.


The problem considered is also interesting. The experimental section is robust and the only point of criticism is that most of the methods compared are simply variants of the proposed approach.

 My main concern with this paper is, however, that the novelty is very limited. It consists in simply applying a collection of well known methods. There is no specific contribution to the problem besides the model described. In terms of core-Machine Learning, the contribution is very small, hence.



**Time Spent Reviewing:**

1

---

> ### Author Response · Authors · 2021-08-06
> **Clarification on comparisons and novelty**
>
> > The experimental section is robust and the only point of criticism is that most of the methods compared are simply variants of the proposed approach.
>
> One of the variants we compared against, Beta-I, is in fact an existing method [Ji et.al.] as noted in Section 3, 4.6 of our paper. Beyond, Ji et.al., there is no other existing method that addresses our problem, to the best of our knowledge
>
> > My main concern with this paper is, however, that the novelty is very limited. It consists in simply applying a collection of well known methods. There is no specific contribution to the problem besides the model described. In terms of core-Machine Learning, the contribution is very small, hence
>
> A major contribution is the identification and formulation of the problem, as noted by  Reviewer DMhv (“worthwhile contribution to a practical problem of great and increasing importance”). Further, Reviewer eCjm notes: “straightforward GPs do not address this problem, and then proposed Dirichlet regularization and pooling”.  Indeed, we developed our approach starting from known methods, and their straightforward application led to our baselines: Beta-I, BernGP, BetaGP. In section 3.2, 3.3, we note the drawbacks of the existing methods for the task in hand: (1) heteroskedastic Beta observations across arms (2) noisy per-arm supervision. We then proposed BetaGP-SL, which addressed problem (1) and BetaGP-SLP, which addressed both (1) and (2). Our methods outperform straight-forward application of off-the-shelf methods. While the component techniques are familiar, we argue that their combination toward the novel and highly motivated task is our second contribution.
>
> **References**
> Disi Ji, Robert L Logan IV, Padhraic Smyth, and Mark Steyvers. Active bayesian assessment for black-box classifiers.

---

> > ### Comment · Reviewer_U6pf · 2021-08-17
> > **Response to authors' rebuttal**
> >
> > I would like to thank the authors for the extra clarifications provided. I still think that the contribution of the paper is on the low end of the scale. However, I have read also the other reviews and I have increased a bit my score.

---

### Official Review · Reviewer_eCjm · 2021-07-16

**Rating:** 7
**Confidence:** 4

**Summary:**

- This paper proposed to use Gaussian processes in the embedded space of the attribution combinations to estimate the accuracy surface with uncertainty, even when faced with extreme sparsity and heteroscedasticity of labeled instances.
- The paper shows that straightforward GPs do not address this problem, and then proposed Dirichlet regularization and pooling to ease the over-smoothing and over-fitting.
- The proposed method (AAA) also tackles uncertain attribute inference, and applied the model to fractionally assign instances across arms with both labeled data and unlabeled data.

**Main Review:**

Overall the paper is easy to read and well-argued.

\- In MF dataset, 12 out of 39 attributes are hand-picked. The selection was made according to how important they are are gender classification. I am curious if uninformative features are included into the feature space, how will the model perform, will the accuracy surface be smooth along these informative axises? And what are the consideration to remove gender-neutral features?

\+ The comparison against BetaGP and Beta-I in Section 4.5 and 4.6 clearly verified the effectiveness of the proposed methods.

\+ I agree with the limitations mentioned in in the last paragraph, and agree that they can be left to be addressed in future work.

**Time Spent Reviewing:**

2

---

> ### Author Response · Authors · 2021-08-06
> **Response to the question on accuracy surface**
>
> Table 10 in the Appendix shows the list of 12 attributes that we have considered for the MF task. These 12 attributes were chosen for their relevance to the gender classification task, e.g., mustache, beard, blond hair. Several other of these attributes are semantically neutral, e.g. blurry, young, eyeglasses, smiling, wearing hat, chubby.
>
> We expect the accuracy surface to be smooth along neutral attribute dimensions. To corroborate our claim, we show the average Euclidean distance between embeddings of arm pairs that only differ on a particular attribute. For example, if we only have 4 attributes and wish to measure the sensitivity of the kernel embeddings to the first attribute, we estimate:
> sensitivity(attribute 1) = $\mathbb{E}_{x, y, z} \|\| \text{emb}([1, x, y, z]) - \text{emb}([0, x, y, z])\|\|^2$.
>
> The table below shows such sensitivity values for all the attributes in order of decreasing sensitivity. We note from the table: (1) embedding space is sensitive to gender specific attributes and dataset artifacts (e.g., blond hair as observed in Sagawa 19 et.al.) as expected (2) neutral attributes do not affect the embeddings, evident from the lower sensitivity. Since GPs operate in the embedding space, the accuracy surface would be flat over neutral attributes.
>
> |Attribute|Embedding sensitivity|
> |----------|----------------------|
> |No_Beard|0.89060|
> |Blond_Hair|0.75058|
> |Mustache|0.74468|
> |Young|0.60510|
> |Blurry|0.54001|
> |Chubby|0.52718|
> |Brown_Hair|0.52279|
> |Male|0.50773|
> |Wearing_Hat|0.48115|
> |Eyeglasses|0.47324|
> |Smiling|0.42127|
> |Black_Hair|0.33770|
>
> Our estimation technique is robust to the presence of uninformative features and we do not exercise any special considerations to remove them.
>
> **References**
> Shiori Sagawa, Pang Wei Koh, Tatsunori B Hashimoto, and Percy Liang. Distributionally robust neural networks for group shifts: On the importance of regularization for worst-case generalization. arXiv preprint arXiv:1911.08731, 2019.

---

> > ### Comment · Reviewer_eCjm · 2021-08-30
> > **Response to authors' rebuttal**
> >
> > Thanks for providing additional experimental results! I think the results can use more explanation on how they are generated but overall look good.

---

### Official Review · Reviewer_DMhv · 2021-07-16

**Rating:** 7
**Confidence:** 3

**Summary:**

The paper introduce a novel approach to evaluate a black-box classifier and capture it's performance in different regions of the input space. This is done by modelling an accuracy surface over the space of combinations of data attributes. The model is a beta-GP with variational inference. It can also be used for active learning. Experimental evaluation shows that the model compares favourably against competing methods and illustrates the rationale behind its gains.

**Limitations And Societal Impact:**

adequately addressed

**Main Review:**

This is a well written paper and worthwhile contribution to a practical problem of great and increasing importance. I appreciate the detailed appendix and comparisons.

**Time Spent Reviewing:**

2

---

### Decision · Program_Chairs · 2021-09-27

**Decision:**

Accept (Poster)

**Comment:**

The reviewers are in consensus that this submission represents an interesting, well-motivated contribution to a field that I personally find is under-represented and -valued at conferences: model evaluation. The reviewers are slightly concerned around the explanation of some results and how they are presented -- I hope the authors will take these comments to heart and encourage them to incorporate these changes in preparing the camera-ready version of their manuscript.